# Early Risk Factors for Obesity in the First 1000 Days—Relationship with Body Fat and BMI at 2 Years

**DOI:** 10.3390/ijerph18158179

**Published:** 2021-08-02

**Authors:** Mercedes Díaz-Rodríguez, Celia Pérez-Muñoz, Jesús Carretero-Bravo, Catalina Ruíz-Ruíz, Manuel Serrano-Santamaría, Bernardo C. Ferriz-Mas

**Affiliations:** 1Department of Nursing and Physiotherapy, University of Cádiz, 11009 Cádiz, Spain; mercedes.diaz@uca.es (M.D.-R.); celia.perez@uca.es (C.P.-M.); jesus.carretero@uca.es (J.C.-B.); 2Clinic Management Unit (CMU), Andalusian Health System, 11510 Cádiz, Spain; linaruizruiz@hotmail.com (C.R.-R.); manuel.serrano.santamaria@juntadeandalucia.es (M.S.-S.)

**Keywords:** childhood, obesity, early risk factors, first 1000 days

## Abstract

(1) Background: Obesity is defined as an excessive accumulation of body fat. Several early developmental factors have been identified which are associated with an increased risk of childhood obesity and increased adiposity in childhood. The primary objective of the present study is to analyse the effect of various early risk factors on Body Mass Index (BMI) and body fat percentage at 2 years of age. (2) Methods: A prospective cohort study design was used, with the sample consisting of 109 mother-child pairs from whom data were collected between early pregnancy and 2 years old. Adiposity was determined based on skinfold measurements using the Brooks and Siri formulae. Mean comparison tests (Student’s *t*-test and ANOVAs) and multiple linear regression models were used to analyse the relationship between early programming factors and dependent variables. (3) Results: Maternal excess weight during early pregnancy (*β* = 0.203, *p* = 0.026), gestational smoking (*β* = 0.192, *p* = 0.036), and accelerated weight gain in the first 2 years (*β* = − 0.269, *p* = 0.004) were significantly associated with high body fat percentage. Pre-pregnancy BMI and accelerated weight gain in the first 2 years were associated with high BMI z-score (*β* = 0.174, *p* = 0.047 and *β* = 0.417, *p* = 0.000 respectively). The cumulative effect of these variables resulted in high values compared to the baseline zero-factor group, with significant differences in BMI z-score (*F* = 8.640, *p* = 0.000) and body fat percentage (*F* = 5.402, *p* = 0.002) when three factors were present. (4) Conclusions: The presence of several early risk factors related to obesity in infancy was significantly associated with higher BMI z-score and body fat percentage at 2 years of age. The presence of more than one of these variables was also associated with higher adiposity at 2 years of age. Early prevention strategies should address as many of these factors as possible.

## 1. Introduction

Childhood obesity is a public health problem recognised as an epidemic illness by the WHO [1]. Obesity is defined as an abnormal or excessive accumulation of fat that may be detrimental to health [2]. Different procedures can quantify adipose tissue, most of which are difficult to perform in clinical practice. Anthropometry stands out for its ease of application and low cost and has been the most widely used technique in clinical and epidemiological studies. Body mass index (BMI) has established itself as the most widely used, practical, easy to apply, inexpensive, and non-invasive anthropometric indicator for classifying overweightness and obesity. As an indicator of body mass relative to height, it has limitations in differentiating body fat from lean mass. Despite this, high BMI values are a good indicator of excess fat [3]. High BMI is associated with risk factors for cardiovascular disease [4]. Another anthropometric indicator is skinfold measurement. It allows the level of subcutaneous fat to be estimated, but does not do so for visceral fat. Using skinfold measurements in several areas and applying specific equations, body density and body fat percentage can be calculated [5]. Although it has limitations, if conducted by trained personnel, it is a suitable method for estimating body fat with a strong correlation with electrical bioimpedance in children [6].

Despite the development of specific strategies and programmes to tackle it, the prevalence of childhood obesity remains high [7]. In Spain, there has been a significant decrease in overweightness in children aged 7–8 years, while obesity has remained stable in both sexes [8]. In order to develop more effective prevention strategies, a change in approach is needed [9]. Research on the role of various environmental factors in the earliest stages of development, which corresponds to the first critical period related to childhood obesity, has been fundamental [10]. Thus, the concept of early programming [11] has acquired a relevant role in childhood obesity prevention because it offers the possibility to plan and develop interventions at the earliest stages of development when prevention strategies are most effective.

Factors which act during this early stage of development, or early risk factors, have been linked to an increased risk of childhood obesity [12]. Depending on their effect on early programming, they can be grouped into pre- and postnatal. Prenatal factors include maternal obesity during early pregnancy, excessive weight gain during pregnancy [13], smoking, gestational diabetes, and caesarean delivery [14]. Postnatal factors include absent or brief breastfeeding [15], accelerated weight gain, excessive milk protein intake [16], and early introduction of complementary feeding.

Often one or more early risk factors appear in the same individual. Several studies have analysed their cumulative effect, finding that the appearance of two or more of these factors in an individual was associated with a progressive increased risk of obesity at 6 years [17], 7–10 years [18], or 4 and 6 years [19] regardless of which factor it was. This association with obesity risk was made through BMI. The cumulative effect on direct indicators of adiposity (DEXA) at 4 and 6 years of age was also analysed [19] and results show a similar clear gradual increase at both ages.

High adiposity at birth appears to be a risk factor for childhood obesity at 5 years of age [20]. It could also be a marker of the effect of prenatal programming, and therefore a good indicator of the effectiveness of interventions developed for use during pregnancy. In the same vein, body fat percentage at 2 years could be considered both an indicator of risk of childhood obesity and a marker of the effect of early programming at the end of the first critical period. We have not found any studies analysing the effect of early risk factors on adiposity at age 2.

The main objective of this study is to analyse the effect of several early risk factors on BMI and body fat percentage at 2 years.

## 2. Materials and Methods

### 2.1. Design

An observational, analytical prospective cohort study design was used. Data were collected on risk factors for obesity in the first 1000 days and anthropometric variables (BMI and skinfolds) at 2 years.

### 2.2. Population

The target population is two-year-old children in the Basic Health Area of Puerto Real, located in the south of Andalusia, Spain, and consisting of two Clinical Management Units: Puerto Real, with four paediatric quotas, and Casines, with two. There are no significant socioeconomic differences between these populations.

The study population is the control group of a more extensive study, described elsewhere with detail [21], whose main objective is to evaluate the effectiveness of a combined educational and care intervention, implemented during the first 1000 days of life, to prevent childhood obesity. They are two-year-olds who attended the child health visit of the year from February 2017 and attended the two-year visit until February 2019.

Individuals were incorporated into the study at the health check they had after the first year of life. All one-year-old infants whose gestation and health controls had been carried out at the Puerto Real Clinical Management Unit, who did not present with any pathology that could significantly affect their growth, and whose parents authorised the mother-child pair’s participation in the study in writing were included.

### 2.3. Sample

Based on the historical data of the Puerto Real Basic Health Area, each paediatric group was estimated to have, on average, 50 births per year. Of the six paediatric quotas, two were selected based on the criteria of being assigned to paediatricians with job stability during the project’s development and who agreed to participate in the study. Due to the decrease in the birth rate in the previous year, it was decided at the beginning of the study to extend the inclusion period until the 2-year health check-ups in July 2019, five months longer than the initial limit. A total of 135 mother-child pairs were included in the study. As can be seen in Figure 1, during the follow-up period, some cases were lost for a variety of reasons (change of address, change of paediatrician, non-attendance at the 2-year health check-up, or difficulties measuring skinfolds), leaving 109 remaining pairs which were analysed.

### 2.4. Study Variables

Independent variables related to the mother-child pair were measured. The variables measured from the mother were: age at delivery; BMI (kg/m^2^) at the start of pregnancy, calculated at the first visit (week 4–7); nutritional status, classified as obese (≥30), overweight (25–29.9), average weight (18.5–24.9), or underweight (<18.5); weight gain during pregnancy, calculated as the difference in weight between the first visit (week 4–7) and either the last visit (week 38) or the weight before delivery, classified as adequate or excessive according to the values recommended in 2009 by the Institute of Medicine (IOM) [22]; smoking during pregnancy, considering her to be a smoker if she smoked any amount of tobacco; gestational diabetes mellitus, measured by asking the mother directly; and type of delivery, categorised as by Caesarean section or not.

Variables measured in the child’s first 2 years were: sex; weight (in grams) at birth and at 4, 6, 12, and 24 months; length (in cm) of the newborn; duration of exclusive breastfeeding (EBF) categorised as ≥4 months (full months) or <4 months; duration of breastfeeding (BF) categorised as up to 12 months (full months) or greater than 12 months, in both cases (EBF and BF) using WHO definitions [23]; accelerated weight gain, assessed as an increase in weight z-score, adjusted for age and sex, at birth and at 4, 6, 12 and 24 months, using the Anthro programme [24], with accelerated weight gain being considered to be an increase in weight z-score greater than +0.67 between birth and 4, 6, 12 and 24 months [25]; and early introduction of complementary feeding (before 4 months).

The following dependent variables were measured at 2 years of age: BMI (kg/m^2^); body fat percentage, calculated using Siri’s formula [26]; and body density calculated from the measurement of the four skinfolds (biceps, triceps, subscapular, and suprailiac), applying Brook’s formulae which are validated for children aged 1 to 11 [27].

### 2.5. Data Collection

Data were collected through a data collection sheet, designed for the study, from two sources: (a) direct questions to parents at the first and second year of life visits by paediatricians, and (b) collected from the records of the integrated health care management system of Andalusia Diraya, both for the newborn and the mother during pregnancy (weight, height, and BMI were the variables collected in this way). Data collection started when the intervention started in the experimental group of pregnant women, so that neither the mothers during pregnancy nor the babies after birth could be influenced by the intervention applied to the experimental group.

The weight and height of the pregnant women were measured with an ADE scale (GmbH & Co, Hamburg, Germany) model M304641-01 with a reading range of 2 to 250 kg and a precision of 100 g and with an ADE stadiometer, attached to the scale, model MZ10023-1 with a reading range of 60–210 cm. Weight at birth and at 4, 6, 12, and 24 months was determined with the infant naked using a Soehnle baby scale, with a reading range of 0–20 kg and a precision of 10 g. The length was measured with the infant supine with an Añó-Sayol rigid stadiometer with a reading range of 25 to 90 cm and a precision of 0.5 cm. BMI at 2 years of age was calculated using the WHO growth standards tables [28]. The z-scores for weight and BMI were calculated using the Anthro software [24].

Skinfolds were measured with a Holtain Skinfold Caliper (Holtain Ltd., Dyfed, UK) with an amplitude of 0–46 mm, a graduation of 0.2 mm, and a constant pressure of 10 g/mm^2^. They were measured on the left side of the body, in triplicate, and by a single observer for each paediatric quota, according to the techniques recommended by the WHO [29].

### 2.6. Statistical Analysis

Description statistics were calculated using the most common summary statistics (means, deviations, and confidence intervals for quantitative variables; frequencies and percentages for qualitative variables).

Data on anthropometric measures (BMI and body fat percentage) were tested for normality with the Kolmogorov-Smirnov test when the sample size of the group was greater than 50 and with the Shapiro-Wilk test otherwise. Gender differences in BMI and body fat percentage were evaluated using the appropriate mean comparison test (Student’s *t*-test when data were normal and the Mann Whitney *U*-test as a non-parametric test for when they were not).

Each factor was categorised according to whether or not it was a risk factor. The factors were included as predictors in multiple linear regression models where the dependent variables were BMI and body fat percentage. The stepwise method was used to introduce variables into the regression. In the case of variables in which several categories could be established, the means for the dependent variable in each group were compared using ANOVA tests when data were normal and the Kruskal-Wallis test otherwise. For the post-hoc analysis of the ANOVA test, Tukey’s test was used. In quantitative factors, the correlations between the two variables were calculated using Pearson’s correlation coefficient (*r*).

Individually significant risk factors were analysed together. The cumulative effect of these factors was analysed by comparing group means using the aforementioned mean comparison tests. Gender differences in the cumulative effect of these factors were also evaluated. For comparing frequencies, the chi-square test was used, trying to avoid categories with counts of less than five.

The statistical program SPSS (version 24) was used to carry out this analysis. A *p*-value of 0.05 or less was considered to be statistically significant.

### 2.7. Ethics

The trial poses little or no risk to participants and their offspring. Mothers gave written consent for the use of their data. A parent or legal guardian gave written informed consent on behalf of their children; at the same time, they are provided with an information sheet where they are exposed to the right to withdraw from the research as well as to request more detailed information about the research, the treatment of the data, or the results obtained. The rights, privacy, and integrity of the participants are guaranteed. The research was conducted following the precautionary principle to prevent and avoid risks to life and health, observing compliance with the recommendations set out in the Declaration of Helsinki. Approval was obtained from the regional Ethics Committee (Comité Coordinador de Ética de la Investigación Biomédica de Andalucía, CCEIBA).

## 3. Results

### 3.1. Descriptive Statistics

The characteristics of the study population are shown in Table 1. The mean BMI value at the beginning of gestation was 26.52, within the overweight range.

Descriptive results for variables considered early risk factors for childhood obesity are shown in Table 2 (maternal/gestational and postnatal variables).

More than half of the women (54.13%) started pregnancy with excess weight (overweightness and obesity). One-third of pregnant women (33.03%) had excessive weight gain. Excess weight gain was more frequent among mothers with excess weight at the beginning of pregnancy (47.46% vs. 16%, χ^2^ = 12.108, *p* = 0.000). There was a low incidence of smoking in pregnancy (7.34%).

21.10% of mothers did not initiate breastfeeding, or breastfeeding duration was less than one month. Once breastfeeding was established, the average duration of breastfeeding was 7.56 months. Only in one case (0.92%) was complementary feeding introduced before four months, indicating that this age limit is well established in the population. The z-score for birth weight was +0.02 SD. It increased from the fourth and especially from the sixth month of life, reaching +0.49 SD at 2 years of age.

The percentage of body fat was significantly higher in boys than in girls, contrary to what would be expected. 66.67% of boys (*n* = 34) and 24.14% of girls (*n* = 14) exceeded the Fomon reference values [30].

### 3.2. Relationship between Independent/Dependent Variables

Each variable considered an early risk factor was categorised according to whether its value was considered risky or not. It was included in a multiple linear regression model, taking BMI and body fat percentage at 2 years as dependent variables. The results are shown in Table 3. For BMI at 2 years of age, the z-score provided by the WHO was used, and among the accelerated weight gains in the 2 years of life, the 0 to 24 months was taken as the one with the most significant variation.

Both excessive maternal weight gain during pregnancy and accelerated infant weight gain between birth and 2 years of age had a statistically significant association with a higher BMI z-score (*p* = 0.047 and *p* = 0.000, respectively) to a higher BMI z-score. Children born to mothers who were overweight at the start of pregnancy had a higher BMI z-score at 2 years (+0.34 SD) than children born to regular weight/underweight mothers (+0.03 SD). However, the differences were not significant (*p* = 0.305). No significant association was found between BMI z-score at 2 years and the type of delivery, gestational diabetes, or a duration of breastfeeding of more than 6 months.

Body fat percentage at 2 years showed a significant relationship with pre-pregnancy BMI, with higher values in the group with BMI ≥ 25 (*p* = 0.026), with maternal smoking during pregnancy (*p* = 0.036), and with accelerated weight gain between birth and 2 years (*p* = 0.004). No association was found with the other variables analysed.

The relationship between the two dependent variables was also analysed, and a highly significant correlation was found between BMI z-score and body fat percentage (r = 0.539, *p* = 0.000).

### 3.3. The Cumulative Effect of Early Risk Factors

Finally, the cumulative effect of the significant factors (maternal BMI before pregnancy, excessive weight gain during pregnancy, and accelerated weight gain between 0–2 years) on the dependent variables was analysed. Smoking was discarded in this part because of the small number of participants in the risk group (*n* = 8). The results are shown in Figure 2.

An ANOVA test for difference of means between groups showed significant differences in BMI z-score and body fat percentage (F = 8.640, *p* = 0.000 and F = 5.402, *p* = 0.002 respectively). To test for differences between groups, the baseline value for both variables was considered to be that of the group that did not accumulate any risk factors. The presence of a single factor changed the values very little. However, the accumulation of two or three factors in the same individual was related to a progressive increase in both variables, with significant differences from the baseline group when three were accumulated. The mean value of the BMI z-scores when two and three factors were accumulated (+0.47 SD and +1.20 SD, respectively) was below the cut-off values identifying overweightness at 2 years (+2 SD).

Table 4 shows the distribution of the number of cumulative early risk factors by sex. Factors with a significant association with body fat percentage and/or BMI z-score were counted as risk factors, as was a duration of breastfeeding of less than 6 months; this has well-established links to childhood obesity in the literature. It is observed that boys accumulated more factors than girls, and this difference was significant.

## 4. Discussion

A very high prevalence of maternal overweightness during early pregnancy (54.13%) was found compared to studies conducted in Spain in 2011 (27.11%) [31]. Some authors have explained maternal overweightness as increasing risk of childhood obesity via a higher birth weight [32]. The multicentre IDEFICS study found a linear relationship between birth weight and childhood obesity, but this relationship disappeared after adjusting for lean mass [13]. Maternal BMI ≥ 25 has a strong correlation with neonatal adiposity [33], and higher adiposity at birth appears as a risk factor for childhood obesity at age 5 years [20]. Thus, the relationship between high maternal BMI during early pregnancy and an increased risk of obesity further down the line may not be so much due to birth weight as due to increased newborn adiposity. In this regard, our results showed a significant positive association between maternal excess weight and body fat percentage at 2 years, but not with BMI z-score at 2 years. In addition to genetic mechanisms and those associated with growing up in an obesogenic family environment, maternal BMI may influence the risk of obesity and, above all, cardiometabolic risk through direct intrauterine mechanisms [34] related to early programming.

Excessive weight gain during pregnancy has been associated with an elevated risk of childhood overweightness/obesity [35]. This effect occurs mainly in the first and second trimesters of pregnancy [36] and is associated with both BMI z-score and several indicators of adiposity [37,38]. Our study did not evaluate weight gain in absolute values, but whether it was adequate or excessive according to previous maternal BMI, as recommended by the IOM [22]. The percentage of pregnant women with excessive weight gain was high (33.03%) when compared with other studies published in Spain (21.8%) [17]. We found a significant positive association between excessive gestational weight gain and BMI z-score at 2 years, in agreement with most studies. Some studies find no association with an increased risk of obesity at six years [17]. The significantly higher frequency of excess gestational weight gain in pregnant women with excess weight at the start of pregnancy may partly explain its association with BMI z-score at 2 years.

Smoking in pregnancy is an independent risk factor for childhood overweightness and obesity [13,39]. The effects of tobacco smoke tend to manifest themselves before the age of 5 years [40] and might even be expected to be found as early as 2 years of age. Gestational smoking did not significantly influence BMI z-score. However, it influenced body fat percentage at 2 years, finding that children of smoking mothers had a higher body fat percentage at 2 years. The low percentage of mothers who smoked (7.34%), lower than that published in other studies (19.0%) [17], may have influenced the assessment of the effect of smoking during pregnancy on adiposity and BMI at 2 years of life.

Gestational diabetes is associated with a higher percentage of body fat without a corresponding increase in lean mass in newborns [41], similar to maternal BMI ≥ 25 [33]. No significant relationship was found with adiposity or BMI z-score at 2 years. Other authors also found no association between gestational diabetes and excess weight (BMI > p85) at 2 years [42], nor with mean BMI at 26 months, but did find an association at later ages, especially between 11–13 years [43].

Most studies show similar results for BMI z-score at 5 years [44] and for obesity risk at 6 years [28], although there are disparate results in the literature [45].

Accelerated weight gain showed a positive association with both body fat percentage and BMI z-score at 2 years. Different cut-off values have been published to define the concept of accelerated weight gain (>+0.67 SD, >+1 SD, >+2 SD) [25,46], although the most commonly used is the weight z-score increase +0.67 SD, which was the cut-off value used in our study. Use of different cut-off values did not change the association between accelerated weight gain and the risk of childhood overweightness or obesity [46]. The period used to calculate weight gain from birth is also variable in the literature. Associations have been found between an increased risk of childhood obesity and accelerated weight gain from birth to 4 months [47], 6 months [17], 1 year, and 2 years [48], and between birth and any of the usual controls in which weight is measured in the first 2 years [46]. The prevalence of accelerated weight gain between 0–2 years observed in our study (44.95%) was an intermediate value compared to those published in other studies with the same criteria for defining the variable, which ranged between 30.7% [49] and 61.7% [46].

This factor is a significant predictor of the risk of childhood obesity. A meta-analysis [48] found that the inclusion of accelerated weight gain from birth to one year of age improved the predictive ability of a previous model without this variable for childhood obesity. This was even greater when weight gain between birth and 2 years of age was used. Our results showed concordant results. Of all the periods analysed, a significant positive association was found between accelerated weight gain between 0–12 months and BMI z-score and even stronger associations between accelerated weight gain between 0–24 months and both of those variables was found. It has been published that accelerated weight gain at some point between 0–2 years has a strong association (*p* = 0.002) with body fat percentage at 5 years [46], suggesting that the mechanism by which this risk factor is associated with childhood obesity would be an excessive accumulation of fat tissue.

Type or duration of breastfeeding was not significantly associated with the variables at 2 years. However, the most significant difference in BMI z-score values was observed when the duration of breastfeeding was between 6–12 months and less than 6 months (0.05 ± 1.04 vs. 0.35 ± 1.02, *p* = 0.134, respectively). Numerous studies support the protective effect of breastfeeding against childhood obesity if breastfeeding is prolonged beyond 6 months [13,42]. This protective effect, however, is disputed. With more excellent protection, the results in favour of the longer the duration of breastfeeding [15] may be due, in part, to other confounding factors: maternal obesity, smoking, or educational level [17]. Exclusive breastfeeding for 4 months or more was also unrelated to the variables at 2 years. A duration of fewer than 4 months has been associated with an increased risk of overweightness/obesity at 1 year of life [50] but is not found at 6 years [17].

The higher body fat percentage value at 2 years in boys was not the expected result. It could be due to the higher accumulation of early risk factors in boys than girls, a difference that was statistically significant, or a higher percentage of children with body fat values above the mean value by sex published by Fomon [30].

When analysing the effect of the accumulation of early risk factors, i.e., with factors that showed a significant association with the outcome variables at 2 years, it is observed that the presence of a single factor barely modified the value of body fat percentage or BMI z-score; however, the association of two or three factors produced a progressive increase in the outcome variables, a relationship which was significant when three risk factors were associated.

This cumulative effect is similar to that found in the risk of childhood obesity as calculated by BMI in several studies, irrespective of which factors they were. Thus, the presence of only one factor did not increase the risk, [19] or only increased it slightly [18], but from two onwards, the risk of childhood obesity increased progressively, the difference being observed either at 6 years (not at 2 and 4 years) [17] or at 4 and 6 years [19]. This similar effect of factor accumulation may indicate that at 2 years of age, both BMI z-score and body fat percentage may be early indicators of obesity risk at later ages.

It has been published that the cumulative effect of more than one early risk factor produces a similar progressive and significant increase also on direct indicators of adiposity (DEXA) at 4 and 6 years of age [19]. Our study finds this effect of the accumulation of risk factors on body fat percentage to already be present at 2 years. To the best of our knowledge, there are no sources in the literature where this relationship this found to be present at 2 years of age.

We found that the mean BMI z-score value at 2 years of age, when two or three factors are present (+0.47 SD and +1.20 SD, respectively), was below the cut-off values for identifying overweightness (+2 SD) and obesity (+3 SD) at this age. Below the age of 5 years, when the values defining overweightness and obesity are broader, adiposity indicators could more accurately define the population at risk of childhood obesity, even from the age of 2 years.

## 5. Conclusions

Maternal excess weight during early pregnancy accelerated weight gain in the first 2 years, and maternal smoking was significantly associated with higher body fat percentage values; excessive weight gain during pregnancy and accelerated weight gain in the first 2 years were associated with higher BMI z-score at 2 years.

The cumulative effect of early risk factors for obesity on BMI and body fat percentage is already observed at the age of 2 years, similar to obesity risk in childhood.

On the one hand, the high prevalence of maternal overweight in early pregnancy should be considered a public health problem and requires developing specific programmes for its prevention. Health interventions focused on preventing childhood obesity that begins early in pregnancy do not address this risk factor. After birth, it is essential to avoid accelerated weight gain during the first 2 years of life, not only in the first months. On the other hand, early prevention strategies should address as many early risk factors as possible.

Future research should determine which of the two variables, at 2 years, best reflects the effect of early programming on obesity risk and test their validity as indicators of the effectiveness of obesity prevention interventions implemented in the first 1000 days.

## 6. Strength, Weakness and Limitation of the Study

The sample size carries limited statistical power. Some of the data used are secondary data, with limitations derived from having been collected for clinical or administrative purposes, but not for research. However, they were collected by the same health personnel who later participated in the study. Losses to follow up reached 20%. Although we cannot rule out that there are differential characteristics between the losses and the participants who remained in the study, we consider that they could hardly affect the quality of the results. The data of some variables were obtained by direct interview and are subject to information biases, although we consider them little relevant in their impact on the results since the parents’ recall of the data related to this stage has proven to be reliable. However, these types of studies in the first thousand days of life using the concept of early programming are not very widespread, so it is necessary to demonstrate the importance of the influence of scheduling factors on the risk of future obesity.

## Figures and Tables

**Figure 1 ijerph-18-08179-f001:**
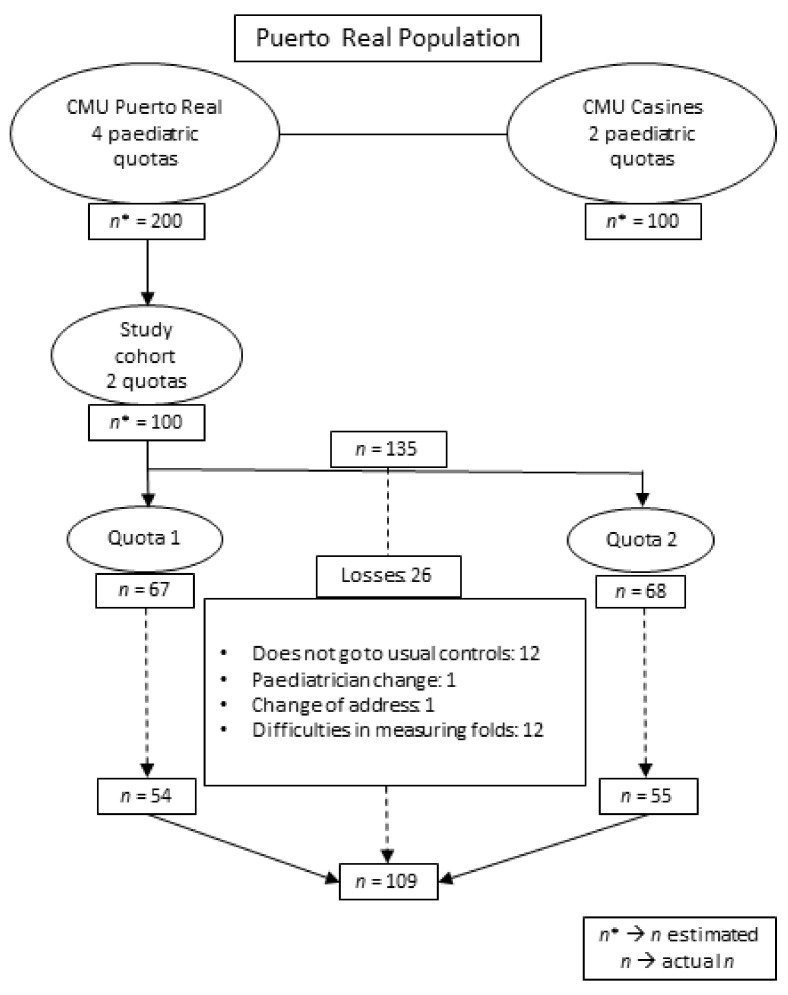
Characteristics of the study population (*n* = 109)*. n** is referred to the initial estimation of *n*.

**Figure 2 ijerph-18-08179-f002:**
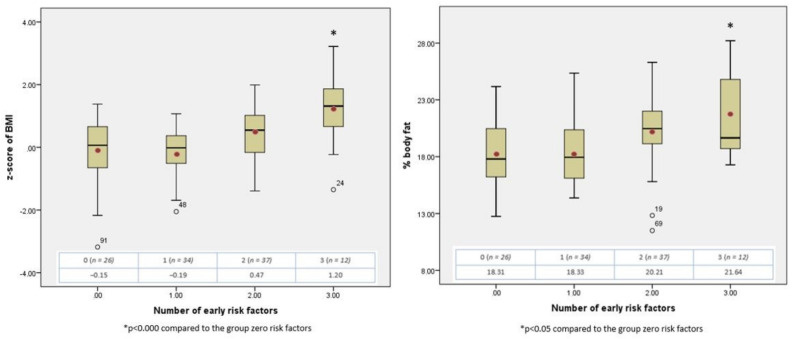
Cumulative effect of early risk factors on BMI z-score and body fat percentage at 2 years. (* indicates significance).

**Table 1 ijerph-18-08179-t001:** Characteristics of the study population (*n* = 109).

Maternal Variables		Mean	SD	95% CI
Maternal age (years)		32.37	5.24	(31.38, 33.35)
Parity		1.65	0.67	(1.53, 1.78)
Pre-pregnancy BMI		26.52	6.02	(25.39, 27.66)
**Newborn Variables**		***n***	**(%)**	
Sex	Female	58	53.21	
	Male	51	46.79	
		**Mean**	**SD**	**95% CI**
Weight (kg)		3.31	0.47	(3.23, 3.40)
Length (cm)		49.81	2.29	(49.38, 50.24)

**Table 2 ijerph-18-08179-t002:** Variables associated with early risk factors (*n* = 109).

		*n*	(%)	
Pre-pregnancy BMI	Average weight/Low weight	50	45.87	
	Overweight	34	31.19	
	Obesity	25	22.94	
Weight gain *	Adequate	73	66.97	
	Excessive	36	33.03	
Smoking	Yes	8	7.34	
	No	101	92.66	
Gestational diabetes	Yes	14	12.84	
	No	95	87.16	
Caesarean delivery	Yes	30	27.52	
	No	79	72.48	
Exclusive breastfeeding ≥4 months		50	45.87	
Prevalence of breastfeeding at 6 months		55	50.45	
Duration of breastfeeding	0 months	23	21.10	
	1–6 months	37	33.94	
	>6 months	49	44.95	
Early complementary feeding (<4 m)		1	0.96	
Accelerated weight gain 0–2 years		49	44.95	
		**Mean**	**SD**	**95% CI**
Duration of breastfeeding (months)		7.56	4.33	(6.64, 8.47)
Increment z-score weight 0–4 m		0.17	1.45	(−0.11, 0.44)
Increment z-score weight 0–6 m		0.40	1.40	(0.13, 0.67)
Increment z-score weight 0–12 m		0.41	1.41	(0.15, 0.68)
Increment z-score weight 0–24 m		0.49	1.30	(0.24, 0.73)
BMI Z-score 2 years	Girls	0.25	1.03	(−0.02, 0.52)
	Boys	0.13	1.05	(−0.16, 0.43)
	Total	0.20	1.04	(−0.00, 0.39)
% body fat 2 years	Girls **	18.43	3.38	(17.56, 19.30)
	Boys **	20.35	2.81	(19.58, 21.12)
	Total	19.33	3.26	(18.72, 19.94)

* Applying weight gain correction for twinship on one mother. ** Significant difference in average fat by sex (t = −3.196, *p* = 0.002).

**Table 3 ijerph-18-08179-t003:** Early risk factors predicting BMI z-score at 2 years and percentage fat mass according to multiple linear regression models.

VD1: BMI at 2 Years (z-Score)	No Risk (±SD)	At Risk (±SD)	β	t	*p*-value
Pre-gestational BMI (</≥25)	0.03 ± 0.97	0.34 ± 1.08	0.094	1.031	0.305
Accelerated weight gain during pregnancy (no/yes)	−0.01 ± 0.95	0.40 ± 1.16	0.174	2.012	0.047 *
Gestational diabetes (no/yes)	0.16 ± 1.06	0.41 ± 0.89	0.059	0.682	0.497
Caesarean section (no/yes)	0.21 ± 1.07	0.17 ± 0.97	−0.042	−0.480	0.633
Smoking (no/yes)	0.24 ± 1.01	−0.37 ± 1.30	−0.116	−1.341	0.183
Breastfeeding ≥6 m (no/yes)	0.05 ± 1.04	0.35 ± 1.02	0.081	0.929	0.355
Exclusive breastfeeding ≥4 m (no/yes)	0.09 ± 1.15	0.28 ± 0.94	0.049	0.566	0.573
Accelerated weight gain 0–24 months (no/yes)	−0.21 ± 0.95	0.69 ± 0.93	0.417	4.826	0.000 *
**VD2: % fat mass at 2 years of age**	**No risk** **(±SD)**	**At risk** **(±SD)**	**β**	**t**	***p*-value**
Pre-gestational BMI (</≥25)	18.61 ± 2.99	19.94 ± 3.39	0.203	2.256	0.026 *
Accelerated weight gain during pregnancy (no/yes)	18.89 ± 3.19	19.61 ± 3.27	0.148	1.541	0.126
Gestational diabetes (no/yes)	19.46 ± 3.23	18.42 ± 3.49	−0.129	−1.404	0.163
Caesarean section (no/yes)	19.37 ± 3.29	19.21 ± 3.24	−0.049	−0.537	0.592
Smoking (no/yes)	19.17 ± 3.22	21.36 ± 3.37	0.192	2.124	0.036 *
Breastfeeding ≥6 m (no/yes)	19.34 ± 3.30	19.32 ± 3.25	−0.084	−0.915	0.362
Exclusive breastfeeding ≥4 m (no/yes)	19.52 ± 3.63	19.17 ± 2.94	0.081	0.929	0.355
Accelerated weight gain 0–24 months (no/yes)	18.55 ± 2.87	20.28 ± 3.48	0.269	2.986	0.004 *

β = standardised coefficient of the multiple linear regression model. The coefficients without covariate adjustment of the final model calculated through the stepwise method are shown. * indicates significance, *p*-value < 0.05.

**Table 4 ijerph-18-08179-t004:** Number of early irrigation factors accumulated by sex.

Factors	Girls	Children	% Total
No factor	9	6	13.76
1 factor	22	5	24.77
2 factors	15	15	27.52
3 factors	9	20	26.61
4 factors	3	5	7.34
chi2 = 15.591, *p* = 0.004

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
