# Peer review of "Early Risk Factors for Obesity in the First 1000 Days—Relationship with Body Fat and BMI at 2 Years"

_ijerph, 2021, doi:10.3390/ijerph18158179_

Round 1

Reviewer 1 Report

Thank you for the opportunity of reviewing the manuscript “Early risk factors for obesity in the first 1,000 days - relation-2 ship with body fat and BMI at 2 years.”

First of all, the authors have successfully found the research gap, addressed current phenomenon and led a way to future direction of intervention. However, the manuscript can be strengthened by providing a clearer layout to illustrate the research questions and corresponding findings in order to support authors’ statements. With these improvements, I believe the manuscript can be used as a culturally sensitive strategy to improve health intervention and the quality of child health in early age.

There are a few suggestions list below and hopefully they can be of help.

  1. I may have missed but couldn’t find the reason why the authors used both BMI and body fat percentage as dependent variables. If the difference could be clearly stated then the concept of the study could be better explained.
  2. Line 59, 70, 72: The terms like “several factors” “another study” “the three study” are confusing.
  3. Line 78: Adiposity and obesity were used interchangeably in this article. It is suggested that the reason of the usage should be explained.
  4. Line 87: The methods and measurements are solid, but the rights of the research participants are expected to be described in the article.
  5. The layout of the description of the findings can be re-arranged. It was somehow confusing during readying with a lot of information. It is suggested that authors may want to present the findings corresponding to the research questions so the structure can be built-up simultaneously.     
  6. Overall, great findings and information given in this article are nice, but creativity and new perspectives are anticipated. If the structure can be developed and the findings can be addressed accordingly, then the quality of the article will be improved.
  7. Strength, weakness and limitation of can be added to support the solidity of the study.
  8. The writing of the paper may need a little help from professional editing. Some problems of typo, wording choice, and sentencing need careful revisit and adjustment.

Reviewer 2 Report

The manuscript Díaz-Rodríguez aimed to evaluate the early-stage risk factors for obesity of children under age 2. The authors focused on the early risk factors; infant obesity was associated with changes in body fat with a higher BMI z-score at two years of age. Even though the article is described well, few queries need to be answered.

  1. Figure 1 shows n=135. But the text line No. 114 shows 137. Please modify it with the appropriate number.
  2. Table 2, the ‘n’ number in the overweight group is missing.
  3. Please mention the table covariates adjusted in Table 3 multiple linear regression model.
  4. Figure 2 please provide a box plot to know the distribution of data and use the origin from zero.
